# Integrating Genomics and Transcriptomics to Identify Candidate Genes for Egg Production in Taihe Black-Bone Silky Fowls (*Gallus gallus domesticus Brisson*)

**DOI:** 10.3390/ijms25179373

**Published:** 2024-08-29

**Authors:** Yuting Tan, Xuan Huang, Chunhui Xu, Yunyan Huang, Shibao Li, Zhaozheng Yin

**Affiliations:** College of Animal Science, Zhejiang University, Hangzhou 310058, China; 22217002@zju.edu.cn (Y.T.); 22017085@zju.edu.cn (X.H.); 12217018@zju.edu.cn (C.X.); 22217090@zju.edu.cn (Y.H.); 22317074@zju.edu.cn (S.L.)

**Keywords:** egg-laying performance, selection signal analysis, RNA-seq, ovarian development

## Abstract

The Taihe Black-Bone Silky Fowl (*Gallus gallus domesticus Brisson*) possesses significant value in terms of consumption, medicinal applications, and ornamental appeal, representing a precious genetic resource and traditional Chinese medicinal material. However, considerable variation exists within populations regarding egg-laying performance. This study integrates a whole-genome selection signal analysis (SSA) with a transcriptome analysis to identify genes associated with egg-laying traits in Taihe Black-Bone Silky Fowls. We identified 31 candidate genes under selection from the high-yield chicken (HC) and low-yield chicken (LC) groups. Additionally, through RNA-seq analysis, 257 common differentially expressed genes (DEGs) were identified from four comparative groups. Two overlapping genes—*LPL* and *SETBP1*—were found in both the selected gene and DEG lists. These selected genes and DEGs were enriched in pathways related to ovarian development, including the lysosome pathway, the ECM–receptor interaction pathway, the TGF-beta signaling pathway, the Wnt signaling pathway, the PPAR signaling pathway, and the glycerolipid metabolism pathway. These research findings contribute to the breeding of Taihe Black-Bone Silky Fowls with high egg production traits and provide a theoretical foundation for exploring the regulatory mechanisms of avian reproduction.

## 1. Introduction

Taihe Black-Bone Silky Fowls, originating from the northern foothills of Wushan in Taihe County, Jiangxi Province, China, exhibit distinctive features known as the “ten characteristics”: a cluster crown, a tassel on the head, green ears, a beard, silk feathers, five claws, feathered feet, black skin, black flesh, and black bones. Renowned for their high culinary, medicinal, and ornamental value, they represent a precious breed resource and traditional Chinese medicinal material [1,2]. The Taihe Black-Bone Silky Fowl population reared by Xichang Fengxiang Poultry Industry Co., Ltd., Jiangxi, China, exhibits varying egg-laying performances. Certain individuals within this population lay fewer eggs under the same rearing conditions compared to others. This provides an excellent model for studying the genetic basis of egg-laying traits.

Egg production traits represent the foremost concern in hen husbandry, as they directly influence the economic viability of poultry farmers. Therefore, augmenting egg yield stands as a primary objective within poultry breeding programs [3]. In modern poultry farming, egg production traits, including egg number (EN) and egg weight (EW), are pivotal reproductive and economic determinants [4]. Eggs constitute one of the most cost-effective and sustainable sources of animal protein. As a dietary staple, they provide humans with a high-quality protein source due to their high digestibility and balanced amino acid composition. Furthermore, eggs have a relatively low water footprint compared to other animal protein sources, making them an important alternative in the context of global sustainability. With the exponential growth of the global population, the demand for poultry eggs is rising correspondingly. Meeting this escalating demand necessitates significant advancements in hen genetics, nutrition, and husbandry practices. Thus, enhancing egg yield and quality from a genetic standpoint is imperative for the sustained development of the global poultry industry [5,6].

Over the past few decades, advancements in genetic and genomic technologies have played a pivotal role in enhancing the productivity of livestock. With the development of next-generation sequencing and the subsequent reduction in associated costs, there is growing interest in utilizing whole-genome resequencing (WGS) as an alternative to SNP genotyping arrays [7]. Selection signal analysis (SSA), a component of WGS, primarily aims to identify regions within the genome subjected to natural selection, thereby revealing distinct features of selection acting upon the genome. In many instances, selection may augment the degree of differentiation among populations. Inferences drawn regarding selection provide robust tools for functional investigations, such as predicting genomic regions potentially associated with diseases. Detecting selection features in chickens is highly significant in aiding the preservation and enhancement of molecular breeding programs [8,9].

Currently, WGS and SSA have been extensively applied in livestock and poultry research. Li et al. conducted whole-genome resequencing on wild and domestic sheep, identifying genes associated with morphological and agronomic traits [10]. Choi et al. performed a whole-genome resequencing analysis on various pig breeds, revealing several loci potentially linked to economic traits in pigs [11]. Luo et al. conducted whole-genome resequencing on Xiangxi cattle, identifying genomic diversity and selection features [12]. Li et al. conducted whole-genome resequencing on ducks, constructing a single-nucleotide polymorphism genetic map of their genome, characterizing their genetic features, and identifying selection features [13]. Sun et al. identified potential candidate genes for egg-laying traits in ducks through whole-genome sequencing and identified key functional pathways associated with poultry egg production related to differential variation genes, including MAPK signaling, Wnt signaling, melanin biosynthesis, calcium signaling pathways, and *FZD6*, identified as a novel candidate gene for egg-laying traits in ducks [4]. Zhao et al. explored candidate genes for goose egg-laying traits through whole-genome selection feature mining, detecting genes and pathways potentially contributing to egg-laying differences, including the PI3K-Akt signaling pathway (*IGF2*, *COMP*, and *FGFR4*), animal organ morphogenesis (*IGF2* and *CDX4*), and female gonad development (*TGFB2*) [14].

Ovarian transcriptomics has long been an essential tool in studying reproductive traits and has been widely employed in investigating the egg-laying performance of poultry. The ovarian follicles in chickens develop through a continuous and hierarchical process. During egg production, functionally mature ovaries contain hundreds of pre-hierarchical follicles, including small and large white follicles (SWFs and LWFs), small and large yellow follicles (SYFs and LYFs), 5–6 growing pre-ovulatory follicles (classified as F6 or F5, F4, F3, F2, and F1 based on volume), and 2–4 post-ovulatory follicles (POFs) lacking oocytes. These follicles, varying in size and developmental stage, possess unique molecular genetic characteristics and play distinct roles in ovarian growth and development [15,16,17]. Studies suggest that increased egg production in chickens relies on the progression of developmental stages and follicular growth [18]. The growth, development, and function of follicles determine the egg-laying performance of chickens, with the rate and stability of yolk deposition playing a crucial role in influencing egg quality [19].

Small white follicles represent the early stage of follicular development, and understanding the changes in endocrine and cellular signaling processes during this stage aids in comprehending the development and maturation of initial follicles. Small yellow follicles represent the subsequent stage of follicular development and are considered crucial for follicular selection and preparation for ovulation. Hormonal regulation and gene expression undergo significant changes during this stage, influencing whether follicles successfully enter the ovulation sequence [20,21,22]. Transcriptomic studies of these two stages may unveil key genes and pathways affecting the ovulation process and egg production. You et al. identified candidate genes associated with egg production through a transcriptome analysis of ovarian tissues from high-egg-producing White Leghorn hens and low-egg-producing chickens during sexual maturity and peak egg-laying periods [23]. Chang et al. utilized transcriptome sequencing to explore the ovarian transcriptomes of high and low-egg-producing ducks, identifying genes involved in ovarian development regulation, including *NPY*, *CDK1*, and *E2F1* [24]. He et al. conducted a transcriptome analysis on the hypothalamus, pituitary, thyroid, and ovarian stroma of normal egg-producing (NG) and abnormal egg-producing (AG) groups, providing valuable insights into regulatory pathways and key genes in the hypothalamic–pituitary–ovarian (HPO) and hypothalamic–pituitary–thyroid (HPT) axes [25]. Li et al. investigated the transcriptomic dynamics of granulosa cells in chicken follicles across ten key developmental stages, discovering that stage-dependent dynamic gene expression correlates with functional differences during folliculogenesis [15]. Follicles constitute the growth and developmental environment for oocytes. Surrounding follicular cell types such as granulosa cells, zona pellucida cells, and cumulus cells significantly influence oocyte development by secreting a series of hormones and growth factors. Therefore, analyzing the follicular transcriptome may reveal how follicular cells support and regulate oocyte maturation through changes in gene expression [26].

Regarding the combined analysis of genomics and transcriptomics in studying egg-laying traits, only a few studies have been conducted thus far. Liu et al. performed whole-genome resequencing and transcriptome sequencing on WuLong geese with different egg-laying capacities, screening candidate genes associated with egg-laying performance [27]. Cai et al. conducted transcriptome sequencing and whole-genome resequencing on four pendulous-comb (PC) and upright-comb (UC) chickens, identifying potential egg-laying candidate genes [28]. Currently, there remains a limited number of articles utilizing the combined approach of whole-genome resequencing and transcriptome sequencing to study egg-laying traits in chickens, indicating the need for further research in this area.

The present study employs whole-genome resequencing technology to identify genomic variations in Taihe Black-Bone Silky Fowls. We conduct a selection elimination analysis on populations with different egg-laying capacities while integrating transcriptome sequencing of ovarian follicles. The aim is to screen for relevant candidate genes and molecular markers influencing egg-laying traits in Taihe Black-Bone Silky Fowls. This research primarily focuses on providing theoretical support for breeding efforts to improve egg-laying performance. Additionally, the findings may offer insights that could aid in the conservation of genetic resources by informing strategies that balance selective breeding with the maintenance of genetic diversity.

## 2. Results

### 2.1. Phenotypic Statistics and Genomic Variation

To compare the egg-laying performance between HC and LC groups, the number of eggs laid by the selected hen populations from onset of laying to 300 days was measured. The results indicated that the egg production of HC group was higher than that of LC group (Figure 1 and Appendix A).

From a total of 54 chickens, we obtained a cumulative 1706.58 Gb of raw sequencing data, with approximately 31.23 Gb of clean data reads obtained per sample after quality control. The sequencing quality was high, with a Q20 ratio exceeding 97% and a Q30 ratio exceeding 92%. The reference genome size was 1,230,258,557 bp, and the alignment rates of all samples to the reference genome ranged from 98.93% to 99.44%. The average coverage depth of the reference genome (excluding N regions) ranged from 14.17× to 45.89×, with a 1× coverage rate (at least one base covered) of over 91.73% (Appendix A). The alignment results were normal, and the sequencing quality, mapping rate, and average sequencing depth met the requirements for subsequent variant detection and related analysis.

### 2.2. Single-Nucleotide Polymorphism (SNP) Detection

In this study, a total of 9,652,548 SNPs were identified. Based on the SNP annotation results, the majority of SNPs were located in intergenic regions (5,266,201, 54.56%), while the remaining SNPs were distributed in intronic regions (3,933,165, 40.75%), upstream regions of genes (128,438, 1.33%), downstream regions of genes (117,559, 1.22%), and exonic regions (123,560, 1.28%). Additionally, among the SNPs located in exonic regions, the majority were synonymous mutations (87,783, 0.91%) and non-synonymous mutations (35,398, 0.37%), with a small fraction representing exonic stop gains and exonic stop losses (Appendix A).

### 2.3. Genetics of the Population and Selection Signal Analysis (SSA)

The results depicted in the PCA scatter plot indicate that while there is some overlap between the HC group (red points) and the LC group (blue points), the two groups show a degree of separation along the PC1 axis. But overall, both groups display a similar spread across the principal components, without clear clustering (Appendix A). The box plot illustrates the distribution of inbreeding coefficients categorized by egg production levels for both groups (Appendix A). The HC group exhibits a lower median inbreeding coefficient. In contrast, the LC group shows a slightly higher median inbreeding coefficient. The LC group exhibits longer average Runs of Homozygosity (ROHs) compared to the HC group, yet the total ROH length distribution is similar between the two groups (Appendix A).

More windows of selection were present on chromosome 1 and chromosome Z, with selected regions identified based on the top 5% cutoff (Figure 2a and Appendix A). The HC and LC groups display distinct distributions of nucleotide diversity, with the HC group showing a slightly higher median nucleotide diversity than the LC group (Figure 2b). Tajima’s D is employed to detect selection regions in the high and low groups, with thresholds set at ≥3.50 and ≤0.12 based on the 95% confidence interval (Figure 2c,d and Appendix A). Given that a single method may yield false-positive signals of selection, both Fst and θπ values were used to define the top 5% windows. As illustrated in Figure 2e, the red areas represent the top 5% selection scan regions in the HC group, determined through a combined screening of Fst and θπ, encompassing 1107 selected signal windows. Furthermore, gene annotations within these 1107 signal windows in the HC group revealed 406 annotated genes (Appendix A). Similarly, Figure 2f highlights the red areas as the top 5% selection scan regions in the LC group, determined through the same method, including 1037 selected signal windows. Annotation of the genes within these 1037 windows in the LC group identified 332 annotated genes (Appendix A). Among these results, 31 genes were commonly selected in both groups’ selection regions.

### 2.4. Gene Ontology (GO) and Kyoto Encyclopedia of Genes and Genomes (KEGG) Analysis on Genes in the Selected Regions

To further understand the functional aspects of the selected genes in different egg-laying groups of Taihe Black-Bone Silky Fowls, a functional enrichment analysis was conducted for the HC and LC groups (Figure 3a–d and Appendix A). In the GO functional enrichment, terms related to reproduction were enriched, including calcium-mediated signaling, maintenance of protein location, biological regulation, receptor binding, and the Wnt receptor signaling pathway, among others. A KEGG pathway analysis revealed enrichment primarily in pathways related to reproduction, such as the lysosome, MAPK signaling, calcium signaling, ECM–receptor interaction, and Wnt signaling pathways.

### 2.5. Transcriptome Sequencing and Alignment Analysis

As shown in Appendix A, a total of 41,634,726 to 47,162,574 raw reads were obtained from each sample, with 40,900,528 to 46,341,010 clean reads after quality control. All samples exhibited a Q20 value of over 98% and a Q30 value of over 94%, with an average GC content of 50.17%. Additionally, the proportion of reads mapped to the reference genome sequence was over 87% for each sample.

### 2.6. Analysis of Differentially Expressed Genes (DEGs)

Between the high-yield chickens’ small white follicles (HWs) and high-yield chickens’ small yellow follicles (HYs), the low-yield chickens’ small white follicles (LWs) and HWs, the LWs and low-yield chickens’ small yellow follicles (LYs), and the LYs and HYs, we identified 7783, 3646, 2872, and 1676 DEGs, respectively. Among the DEGs between the HWs and HYs, 3957 were upregulated and 3826 were downregulated. For the LWs and HWs, 2116 were upregulated and 1530 were downregulated. In the LW and LY groups, 1518 were upregulated and 1354 were downregulated. Between LYs and HYs, 1030 were upregulated and 646 were downregulated (Figure 4a–e). Across the four comparison groups, 257 genes were differentially expressed in common (Figure 4f).

### 2.7. GO and KEGG Analysis for DEGs

A functional enrichment analysis was conducted for the DEGs identified in the HW vs. HY, LW vs. HW, LW vs. LY, and LY vs. HY comparisons (Appendix A). The GO analysis revealed that differentially expressed genes were predominantly enriched in five categories, including cell–substrate adhesion, skeletal system development, cellular response to growth factor stimulus, extracellular matrix, and collagen binding. The KEGG analysis of differentially expressed mRNAs significantly enriched pathways such as the lysosome, ribosome, focal adhesion, ECM–receptor interaction, TGF-beta signaling, Wnt signaling, GnRH signaling, and steroid biosynthesis pathways (Figure 5a–h). It is evident that a significant portion of the enriched pathways for DEGs are associated with reproduction and share similarities with pathways enriched for selected genes.

### 2.8. Joint Analysis of WGS and RNA-Seq to Explore the Candidate Genes of Egg Production Performance between HCs and LCs

Based on the above results, we conducted a joint analysis of WGS and RNA-Seq data to further refine candidate genes that may influence egg-laying performance in the HC and LC groups. We examined the overlap between selected genes (identified through whole-genome selection signal analysis) and DEGs (obtained from transcriptome sequencing) and identified two candidate genes, *LPL* and *SETBP1* (Figure 6).

### 2.9. Validation of RNA-Seq by Quantitative Real-Time PCR (qRT-PCR)

To validate the RNA-seq data, we selected candidate genes and genes related to follicular development for a qRT-PCR analysis (Figure 7). The results showed that the differentially expressed genes exhibited the same expression trends in both qRT-PCR and RNA-seq, confirming their accuracy.

## 3. Discussion

This study conducted 20× depth WGS on 54 Taihe Black-Bone Silky Fowls with high and low egg production, yielding 1706.58 Gb of high-quality reads. The effective reads were aligned to the chicken reference genome with an average alignment rate of 99.34%, ensuring the quality and reliability of the sequencing results. A total of 9,652,548 SNPs were detected in the Taihe Black-Bone Silky Fowl population. To date, research on the genomic characteristics of Taihe Black-Bone Silky Fowls remains limited. However, the rich data obtained in this study provide a solid foundation for future genetic studies on this breed.

Under consistent nutritional and environmental conditions, differences in egg production levels within the same population may be attributed to genetic factors. There is substantial evidence that specific breeds or genetic lines of chickens have been selectively bred to enhance reproductive traits, including higher egg production [29]. A population genetic analysis revealed distinct distributions of nucleotide diversity between the HC and LC groups, suggesting that genetic variation may play a crucial role in egg production. The higher median nucleotide diversity observed in the HC group could indicate a broader genetic base, potentially offering greater adaptability and resilience to selective pressures. This variability may be associated with more efficient reproductive physiology and overall better egg production traits. Conversely, the lower diversity in the LC group may limit its adaptability and could suggest past selection bottlenecks or less intensive selection for reproductive traits. This finding underscores the importance of maintaining genetic diversity in breeding programs to enhance productivity and adaptability.

Using multiple selection signal methods to detect and assess signals between the Taihe Black-Bone Silky Fowls between the HC and LC groups, 31 common selected genes were found in the selected regions of both groups. A functional enrichment analysis of the selected genes in both groups revealed that GO terms such as calcium-mediated signaling, maintenance of protein location, biological regulation, receptor binding, and the Wnt receptor signaling pathway, as well as KEGG pathways including the lysosome, MAPK signaling, calcium signaling, ECM–receptor interaction, and Wnt signaling pathways, may be associated with egg production.

We collected small white follicles and small yellow follicles from Taihe Black-Bone Silky Fowls with different egg production levels and performed transcriptome sequencing. A total of 514,602,176 high-quality reads were obtained, with an overall alignment rate of over 87% to the chicken reference genome. This successfully constructed 12 transcriptome libraries of Taihe Black-Bone Silky Fowls with excellent sequencing quality, providing abundant transcriptome data for the study of hen egg production performance. The data were annotated to specific genes and subjected to a differential gene expression analysis. DEGs were identified separately for small white follicles and small yellow follicles of chickens with different egg production levels, as well as for those with the same egg production level. This led to the identification of 7783, 3646, 2872, and 1676 DEGs, with the highest number of DEGs observed in the small white and small yellow follicles of high-egg-producing chickens. A GO functional annotation and KEGG pathway enrichment analysis were performed on these DEGs. The results revealed the enrichment of pathways such as the lysosome, ribosome, focal adhesion, ECM–receptor interaction, TGF-beta signaling, Wnt signaling, GnRH signaling, and steroid biosynthesis pathways, which are closely related to poultry reproductive performance, consistent with the enrichment results of genomic selection signals.

Based on our resequencing and transcriptome data, the KEGG pathway analysis revealed several pathways potentially related to egg production, including the lysosome, ECM–receptor interaction, and Wnt signaling pathways. The involvement of the lysosome pathway in both genomic and transcriptomic analyses suggests its role in cellular homeostasis and recycling processes, where efficient lysosomal activity may support the increased turnover of cellular components required for follicle development and oocyte maturation [30]. The ECM–receptor interaction pathway is crucial for maintaining tissue integrity, follicle development, and ovulation [31]. Additionally, during follicle maturation, follicles are exposed to various WNTs that regulate target gene expression through the canonical Wnt pathway, playing a significant role in follicle maturation, ovulation, and luteinization [32].

Finally, a comprehensive analysis of the WGS and RNA-seq data revealed two overlapping genes, *LPL* and *SETBP1*, indicating their potential significance in egg production. Specifically, *LPL* showed significantly higher expression in LWs compared to HWs, higher expression in HYs compared to HWs (*p* < 0.05), and higher expression in LYs compared to HYs, with higher expression in LYs than LWs (*p* < 0.05). Similarly, *SETBP1* exhibited significantly higher expression in HYs compared to HWs, higher expression in LYs compared to LWs (*p* < 0.05), and higher expression in LYs compared to HYs, with higher expression in LWs than HWs (*p* < 0.05). Overall, the *LPL* and *SETBP1* genes are downregulated in the HC group. Additionally, *LPL* was found to be enriched in pathways such as the PPAR signaling pathway and glycerolipid metabolism pathway, which are related to reproduction. A nonsynonymous mutation site was identified within the exons of the *LPL*. This candidate SNP may be associated with egg-laying traits in Taihe Black-Bone Silky Fowls.

Lipoprotein lipase (*LPL*) is one of the most important factors in the distribution and metabolism of lipids throughout the body. It mediates the hydrolysis of triglycerides in lipoproteins such as chylomicrons and very-low-density lipoproteins (VLDLs) in the vascular endothelium. As a key enzyme in lipoprotein metabolism, it is mainly expressed in tissues with a high demand for lipid oxidation or storage [33,34]. Upregulation of *LPL* in the chicken ovary during sexual maturation promotes lipid metabolism and steroidogenesis in granulosa cells (GCs), while inhibition of *LPL* leads to decreased mRNA expression of lipid droplets (LDs). LDs are responsible for various physiological processes, including follicle development, and their content increases gradually with follicle growth [35]. As oogenesis progresses, the normalized transcript abundance of *LPL* in the ovary increases, especially during the period of yolk occurrence, corresponding to a significant increase in ovarian lipid deposition and *LPL* activity [36]. This is consistent with our research results, as in each group, the expression level of *LPL* in small yellow follicles is higher than that in small white follicles. The process of egg laying requires energy and nutrient supply, and lipid metabolism and steroidogenesis are closely related to the egg production of chickens. Stimulating lipid metabolism and steroidogenesis can promote ovarian function, thereby affecting egg production in chickens [37].

SET binding protein 1 (*SETBP1*) encodes a transcription factor (TF) involved in various cellular processes. Research indicates that *SETBP1* is widely expressed, with particularly high expression in the uterus. Its targets are also widely expressed in tissues and function through different pathways by clustering gene expression [38]. *SETBP1* has been identified as a key target of *TRIM29*, and the *SETBP1*/SET/PP2A axis has been shown to be essential for the progression of ovarian cancer driven by *TRIM29* [39]. The abundance of genes regulated by *SETBP1*, especially those associated with multi-organ development, is of particular interest, as their hallmark is the presence of developmental abnormalities in multiple organs. *SETBP1* induces transcriptional networks of developmental genes by acting as an epigenetic hub [40]. Currently, there are no studies directly linking *SETBP1* to egg-laying traits in chickens.

The impact of fat on poultry ovarian development is a relatively novel research area. Milisits et al. [41] found that a lower fat content in the beginning of egg laying in hens is favorable for long-term egg production, suggesting a potential limitation in fat content in hens, beyond which egg production is negatively affected. Studies have shown that a high-fat maternal diet affects the offspring’s ovarian development, ovarian reserve function, and estrous cycle through inducing oxidative stress, apoptosis, and altering the expression of certain genes [42]. High-fat diets influence epigenetic mechanisms, leading to the differential regulation of gene and protein expression due to abnormal methylation levels, which interfere with body weight and oocyte maturation. This results in developmental defects in embryos and poor oocyte quality [43]. However, the precise mechanisms of these effects remain poorly understood. The PPAR signaling pathway indirectly participates in oocyte maturation and ovulation by regulating steroid hormone synthesis in granulosa cells, thereby affecting follicular development and normal ovarian function [44]. Understanding how the PPAR signaling pathway regulates *LPL* expression and function, and consequently impacts lipid metabolism and ovarian development, will provide deeper insights into how lipids affect ovarian health. This could offer important clues for enhancing egg production performance in hens.

## 4. Materials and Methods

### 4.1. Chicken and Samples

From the chicken coops of Xichang Fengxiang Poultry Co., Ltd., Jiangxi, China, fifty-four 300-day-old Taihe Black-Bone Silky Fowls with good growth conditions, normal health status, and complete egg-laying records were selected. These chickens, part of one of the farm’s breeding lines, were managed under consistent environmental and nutritional conditions. The composition of their diet is available in Appendix A. Egg production was meticulously recorded daily from the onset of laying until the age of 300 days. Based on these records, the chickens were categorized into high-egg-producing and low-egg-producing groups. In the sample selection process, based on our recorded egg-laying data and body weight data, we excluded outliers to ensure the rationality of the selection.

To extract DNA, blood was drawn from the wing veins of each chicken and deposited into EDTa-containing tubes, which were then stored at −20 °C. Additionally, six chickens (three high producers and three low producers) were randomly selected and euthanized by cervical dislocation. Ovarian tissues were harvested from these specimens and preserved in liquid nitrogen for RNA extraction.

### 4.2. Whole-Genome Resequencing (WGS) and Quality Control

Genomic DNA (gDNA) was extracted from blood samples using the universal magnetic-bead-based genomic DNA extraction kit (TIANGEN, Beijing, China). The quality of the DNA samples was assessed by agarose gel electrophoresis to evaluate the degree of DNA degradation and the presence of RNA or protein contamination. DNA purity was determined using a Nanodrop spectrophotometer (Thermo Fisher Scientific, Wilmington, DE, USA) (OD260/280 ratio). Accurate DNA concentration measurements were conducted using a Qubit 3.0 fluorometer (Thermo Fisher Scientific, Waltham, MA, USA). Samples with an OD260/280 ratio between 1.8 and 2.0 and a total DNA amount exceeding 1.5 μg were selected for library construction.

The qualified genomic DNA was fragmented into approximately 350 bp pieces using a Covaris sonicator (Covaris, Woburn, MA, USA). The purified DNA fragments were then subjected to end-repair processes and purified using Agencourt AMpure XP magnetic beads (Beckman Coulter, Indianapolis, IN, USA), resulting in 5′-phosphorylated blunt-ended DNA fragments. A single ‘A’ nucleotide was added to the 3′ ends of the double-stranded DNA, following which sequencing adapters were ligated to both ends of the library DNA using T4 DNA ligase. The library was simultaneously size selected and purified using the Agencourt SPRIselect reagent kit (Beckman Coulter, Indianapolis, IN, USA). High-fidelity polymerase was used to amplify the initial library to ensure an adequate total library yield. The PCR products were purified using the AMPure XP system (Beckman Coulter, Beverly, CA, USA). The quality of the library was assessed on the Agilent 5400 system (Agilent Biotechnologies, Palo Alto, CA, USA), and the library was quantified by QPCR (1.5 nM).

Finally, the library was sequenced on an Illumina platform using PE150 sequencing (Illumina Inc., San Diego, CA, USA). The raw image data files generated were processed for base calling to produce raw sequencing reads, which were stored in FASTQ [45] format, containing sequence information and corresponding quality scores. Quality control measures, including raw data filtering and an analysis of sequencing error rate distribution and sequencing data quality, were performed to obtain clean data, which served as the basis for subsequent analyses.

### 4.3. Detection and Annotation of Genomic Variants

The valid sequencing data were aligned to the reference genome using the Burrows–Wheeler Aligner (BWA) (version 0.7.8) [46] with the parameters set to “mem -t 4 -k 32 -M”. The raw mapping results were stored in BAM format. Duplicate results were then removed using SAMtools (version 1.3.1) [47] with the parameter “rmdup” and Picard (available at http://broadinstitute.github.io/picard/, accessed on 1 February 2024). The initial set of SNPs was called using SAMtools (version 1.3.1) with the command “-C 50 -mpileup -m 2 -F 0.002 -d 1000”. These SNPs were subsequently filtered using the following criteria: depth at variant positions > 4 and mapping quality > 20. Functional annotation of the variants was performed using ANNOVAR (version 2015Dec14) [48]. The UCSC Known Genes database was used for gene and region annotations.

### 4.4. Population Analysis and Detection of Selective Signatures

In this study, Taihe Black-Bone Silky Fowls were divided into HC and LC groups, and population SNPs were detected using software such as SAMtools (version 1.3.1), followed by filtering to obtain high-quality SNPs. ANNOVAR (version 2015Dec14) was utilized to annotate the SNP detection results of the population. In the process of conducting a Principal Component Analysis (PCA), we utilized GCTA (version 1.24.2) (with parameters --pca 3 --thread-num 2) to compute the eigenvectors and eigenvalues, followed by plotting the PCA distribution using R (version 4.0.0), where the axes represent different principal components. A PCA effectively simplifies the complexity of multiple correlated variables by linearly transforming them into a fewer number of significant variables. The inbreeding coefficients are calculated using VCFtools (version 0.1.14) (--het) [49], reflecting the degree of inbreeding within the population. These coefficients are instrumental in assessing the genetic diversity, inbreeding pressure, and genetic structure of the population. An analysis of the ROHs for different egg production groups was conducted using VCFtools (version 0.1.14). Several parameters and thresholds were used to define the ROH, including (1) a sliding window of 50 SNPs across the genome; (2) a minimum of 20 consecutive SNPs required; (3) a minimum length of each ROH of 40 kb; (4) an SNP density of one per 50 kb, with a maximum of five missing genotype SNPs allowed per ROH and a maximum of one heterozygous genotype.

We employed a combination of methods including Fst (Fixation Index: a statistical measure of population differentiation), θπ (nucleotide diversity: a statistic for population nucleotide diversity), and Tajima’s D (Tajima’s D Test: a statistic comparing the mean of pairwise differences with the number of segregating sites), along with a combined approach of Fst and θπ, to detect and assess the selective characteristics of the HC and LC groups. Initially, using VCFtools (version 0.1.14) with parameters “--fst-window-size 4000 --fst-window-step 2000”, we calculated the Fst and Pi ratio for different genomic regions between the two groups. We set the parameters (--TajimaD 40000 --step 20000) to compute Tajima’s D values for windows and selected these windows based on θπ, Tajima’s D, or Fst values. From prior research, the threshold for identifying putative selected regions in the θπ, Tajima’s D, and Fst analysis was set at the top 5% window values. Furthermore, a combined analysis using Fst and θπ allowed for mutual validation, with the overlapping regions in the top 5% of these methods being highlighted as selected candidate regions.

### 4.5. Transcriptome Data Comparison and Expression Analysis

Small white follicles (<3 mm) and small yellow follicles (6–8 mm) were collected from the ovaries of three HCs and three LCs. After removing surrounding blood vessels and connective tissues using fine tweezers and a scalpel, the follicles were immediately frozen in liquid nitrogen and stored at −80 °C for RNA extraction. Total RNA was extracted from each sample using TRIzol reagent (Invitrogen, supplied by Sunbio Biotech Co., Ltd., Beijing, China). RNA integrity and the presence of any DNA contamination were verified by agarose gel electrophoresis. The purity and concentration of the total RNA were measured using a NanoDrop spectrophotometer (Thermo Fisher Scientific, Wilmington, DE, USA). RNA integrity was further confirmed using an Agilent 2100 Bioanalyzer (Agilent Technologies, Santa Clara, CA, USA). Samples with an A260/A280 ratio between 1.8 and 2.0 and an RNA Integrity Number (RIN) equal to or greater than 7.0 were considered to be of acceptable quality.

Ribosomal RNA (rRNA) was removed from the total RNA, which was then fragmented into short sequences of 250 to 300 bp. These RNA fragments served as templates for first-strand cDNA synthesis using random oligonucleotides as primers, followed by second-strand cDNA synthesis using a mixture of dNTPs, including dUTP. The resulting double-stranded cDNA was end repaired, A-tailed, and adaptor ligated. cDNA fragments of approximately 350–400 bp were selectively purified using AMPure XP beads (Beckman Coulter, Indianapolis, IN, USA). The second strand containing uracil in the cDNA was degraded using USER enzyme (New England Biolabs, Ipswich, MA, USA). After PCR amplification, 12 libraries were constructed. The libraries were initially quantified using a Qubit 2.0 Fluorometer (Thermo Fisher Scientific, Foster City, CA, USA), diluted to 1.5 ng/µL, and then the insert size of the libraries was assessed using an Agilent 2100 Bioanalyzer (Agilent Technologies, Santa Clara, CA, USA). Once the insert size was confirmed to meet expectations, the library’s effective concentration was accurately quantified by qRT-PCR (effective concentration higher than 1.5 nM) to ensure library quality. Upon passing quality checks, different libraries were pooled according to their effective concentrations and the desired output data volumes for Illumina sequencing.

Initial raw data were filtered to check for sequencing errors and assess GC content distribution, obtaining clean reads for subsequent analyses. Clean reads were aligned to the reference genome using the HISAT2 (version 2.0.5) [50] for rapid and accurate mapping. Novel transcripts were assembled using the StringTie (version 1.3.3) [51]. Gene expression quantification was performed using the featureCounts (version 1.5.0-p3) tool within the Subread (version 2.0.3) [52] based on the alignment of genes to their respective genomic locations, counting the number of reads covering each gene from start to end. After quantification, genes exhibiting significant differential expression between different conditions were identified.

### 4.6. Functional Enrichment Analysis of Selected Genes and Differentially Expressed Genes

The gene set underwent a GO functional enrichment analysis and KEGG pathway enrichment analysis using the clusterProfiler. GO is a comprehensive database describing gene functions, which can be categorized into biological processes, cellular components, and molecular functions. The KEGG, on the other hand, integrates genomic, chemical, and systemic functional information into a comprehensive database.

### 4.7. Joint Analysis of WGS and RNA-Seq

We conducted a cross-comparison of the genes identified as under selection from the WGS data with the DEGs identified from the transcriptome analysis, selecting those genes that showed significant results in both datasets. Further, a literature review was performed on these integrated candidate genes to investigate their potential roles in egg production regulation. By integrating data from different omics approaches, we can more comprehensively identify candidate genes associated with egg production and provide multiple lines of evidence at both the genomic and expression levels, thereby enhancing the reliability and biological significance of our results.

### 4.8. Quantitative Real-Time PCR

Total RNA was extracted from follicular samples using the Trizol method. RNA was reverse transcribed using a reverse transcription kit (TIANGEN, Beijing, China). β-Actin was used as the internal reference. Primer sequences were designed using Primer 5.0, as shown in Table 1, and synthesized by Beijing Tsingke Biotech Co., Ltd. (Beijing, China) qRT-PCR was performed using CFX96 Touch (Bio-Rad Laboratories, Inc., Hercules, CA, USA). The reaction system consisted of 20 µL, including 10 µL of 2× SuperReal PreMix Plus (SYBR Green; TIANGEN, Beijing, China), 0.6 µL of forward primer (10 pmoL/μL), 0.6 µL of reverse primer (10 pmoL/μL), 1 µL of cDNA template, and 7.8 µL of RNase-free water. Each sample was analyzed in three biological replicates and three technical replicates. Gene expression levels were calculated using the −∆∆Ct method. Gene expression bar charts were plotted using GraphPad Prism 9.

### 4.9. Statistical Analyses

Data were analyzed using SPSS version 27.0 (IBM). Initially, the normality of the data distribution was assessed to determine the appropriate statistical tests. For data that followed a normal distribution, significance testing was performed using *t*-tests or a one-way ANOVA. Outliers were identified and removed by excluding data points that deviated more than three standard deviations from the mean. Graphical representations were created using GraphPad Prism 9 software. A *p*-value of <0.05 was considered statistically significant.

## 5. Conclusions

In this study, a whole-genome selection signal analysis was conducted on populations of high- and low-egg-producing Taihe Black-Bone Silky Fowls, identifying 31 genes selected in both groups. Subsequently, transcriptome sequencing was performed, revealing 7783, 3646, 2872, and 1676 DEGs between the high- and low-egg-producing groups in both small white and small yellow follicles, with 257 genes differentially co-expressed in four groups. Additionally, two overlapping genes—*LPL* and *SETBP1*—were found between the selected genes and differentially co-expressed genes. These two genes were downregulated in high-yielding chickens, suggesting their potential role in negatively regulating egg production.

## Figures and Tables

**Figure 1 ijms-25-09373-f001:**
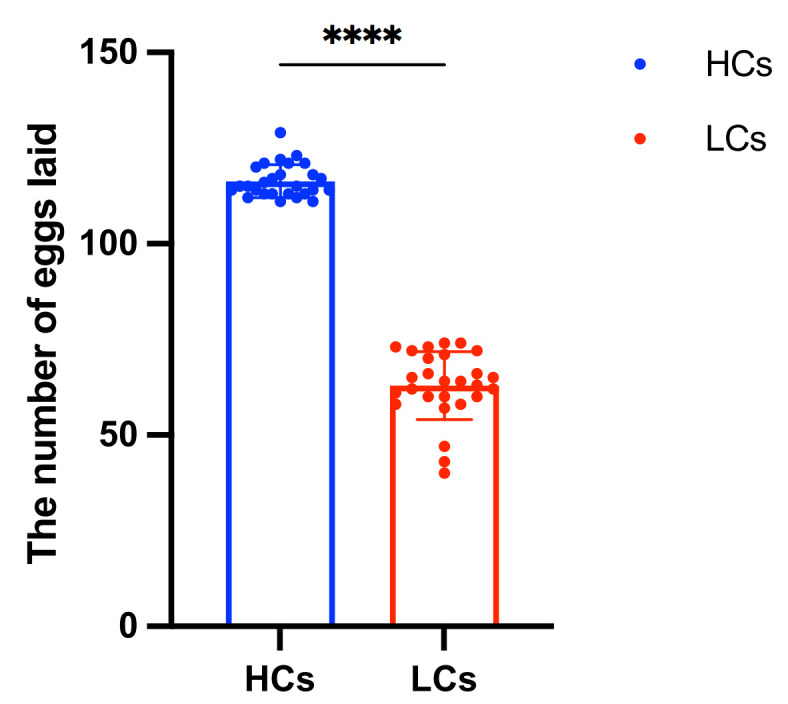
Comparison of the egg-laying number between HC and LC groups. The HCs exhibited a higher number of eggs compared to LCs. The symbol “****” indicates significant differences at *p* < 0.0001. Abbreviations: HCs, high-yield chickens; LCs, low-yield chickens.

**Figure 2 ijms-25-09373-f002:**
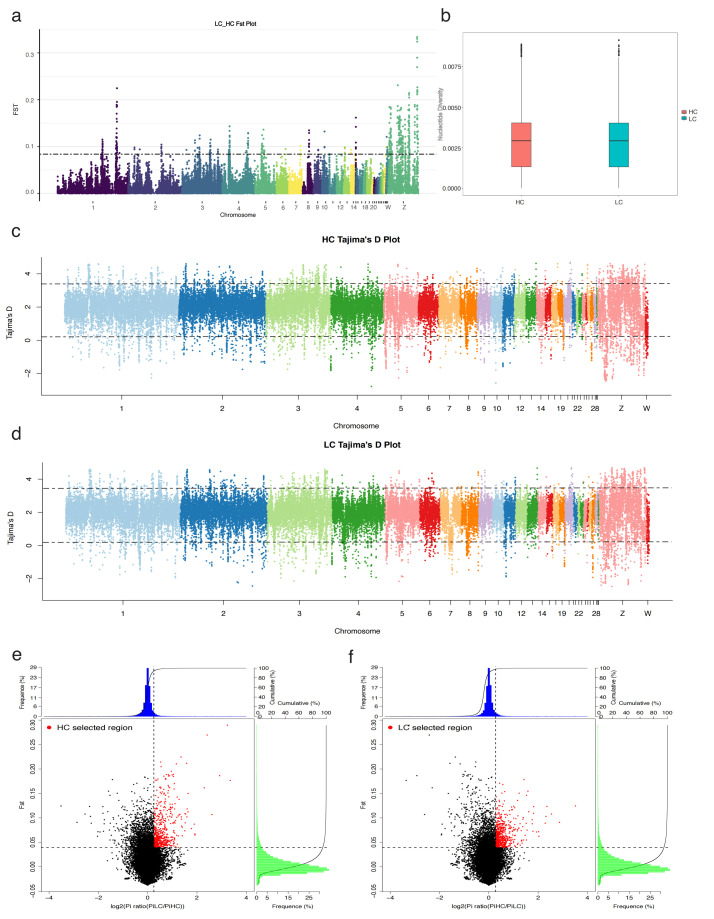
Multiple selection sweep analysis in the HC and LC groups: (**a**) the Fst plot of selective sweeps in the HC and LC groups; (**b**) the box plot of nucleotide diversity for the HC and LC groups; (**c**) Tajima’s D plot of selective sweeps in the HC group; (**d**) Tajima’s D plot of selective sweeps in the LC group; (**e**) Fst and πratio selective elimination analyses of the HC group; (**f**) Fst and πratio selective elimination analyses of the LC group.

**Figure 3 ijms-25-09373-f003:**
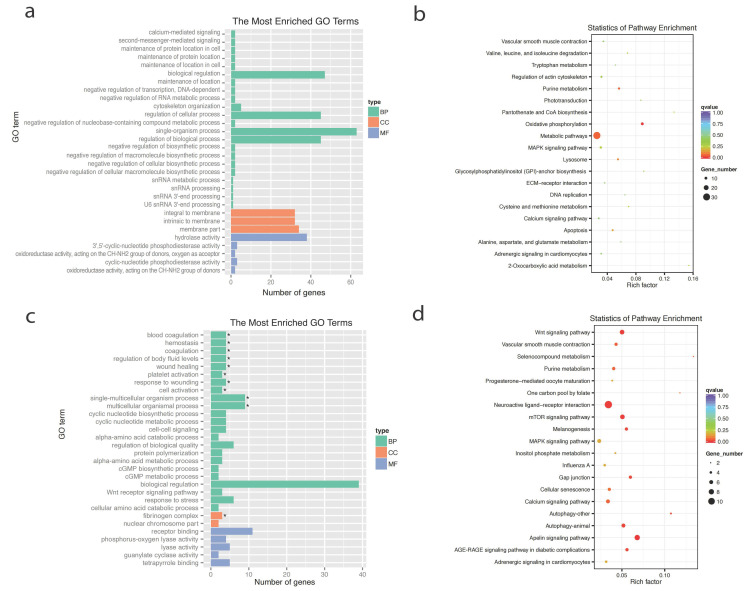
The enrichment analysis of selected genes in the HC and LC groups: (**a**) bar plot of the first 30 GO terms in the HC group; (**b**) bubbles of the top 20 KEGG pathways in the HC group; (**c**) bar plot of the first 30 GO terms in the LC group; (**d**) bubbles of the top 20 KEGG pathways in the LC group. Abbreviations: BP, biological process; CC, cellular component; MF, molecular function. * *p* < 0.05.

**Figure 4 ijms-25-09373-f004:**
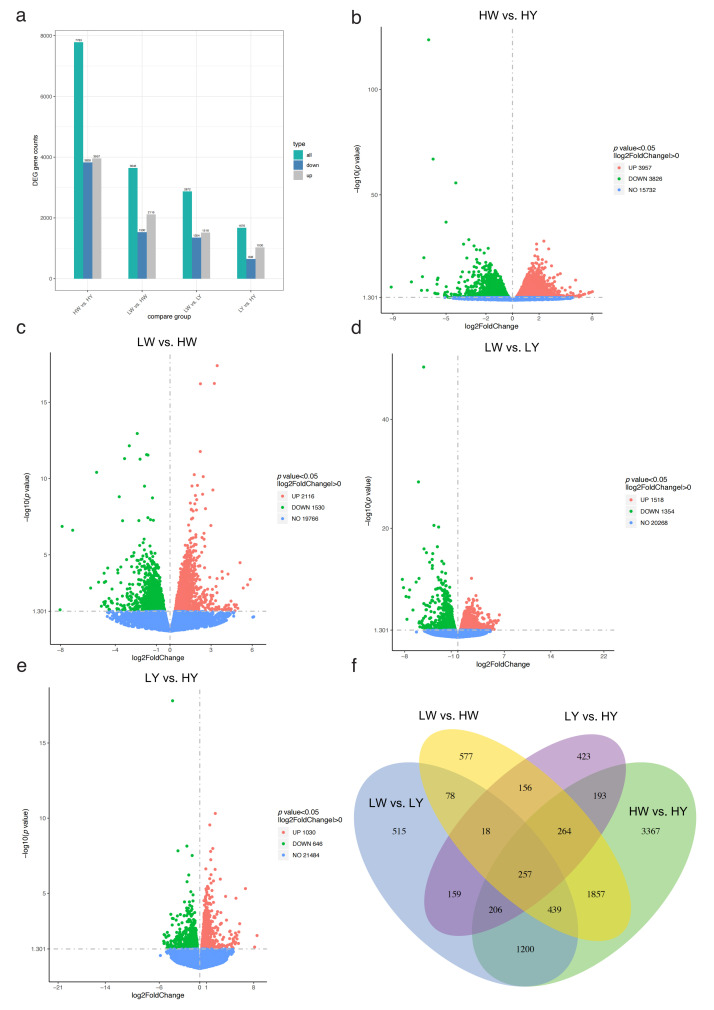
Transcriptome sequencing analysis of HC and LC groups’ ovarian follicles: (**a**) the number of DEGs in the four comparison groups; (**b**) volcano plot for HWs vs. HYs; (**c**) volcano plot for LWs vs. HWs; (**d**) volcano plot for LWs vs. LYs; (**e**) volcano plot for LYs vs. HYs; (**f**) Venn plots of the co-expressed genes among the four comparison groups. The red dots represent significantly upregulated genes, and the green dots represent significantly downregulated genes. Abbreviations: HWs, high-yield chickens’ small white follicles; LWs, low-yield chickens’ small white follicles; HYs, high-yield chickens’ small yellow follicles; LYs, low-yield chickens’ small yellow follicles.

**Figure 5 ijms-25-09373-f005:**
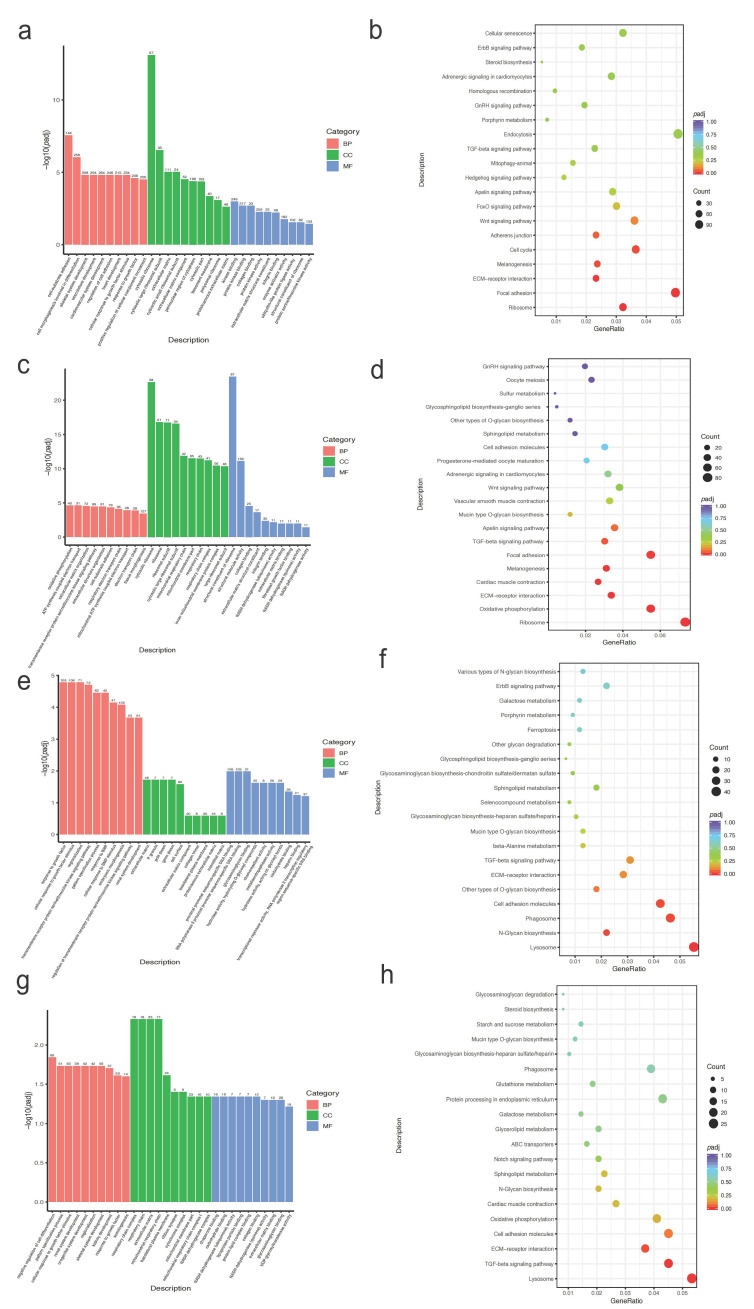
The GO and KEGG enrichment analysis of DEGs in the transcriptome sequencing analysis: (**a**) bar plot of GO terms for HWs vs. HYs; (**b**) bubbles of KEGG pathways for HWs vs. HYs; (**c**) bar plot of GO terms for LWs vs. HWs; (**d**) bubbles of KEGG pathways for LWs vs. HWs; (**e**) bar plot of GO terms for LWs vs. LYs; (**f**) bubbles of KEGG pathways for LWs vs. LYs; (**g**) bar plot of GO terms for LYs vs. HYs; (**h**) bubbles of KEGG pathways for LYs vs. HYs.

**Figure 6 ijms-25-09373-f006:**
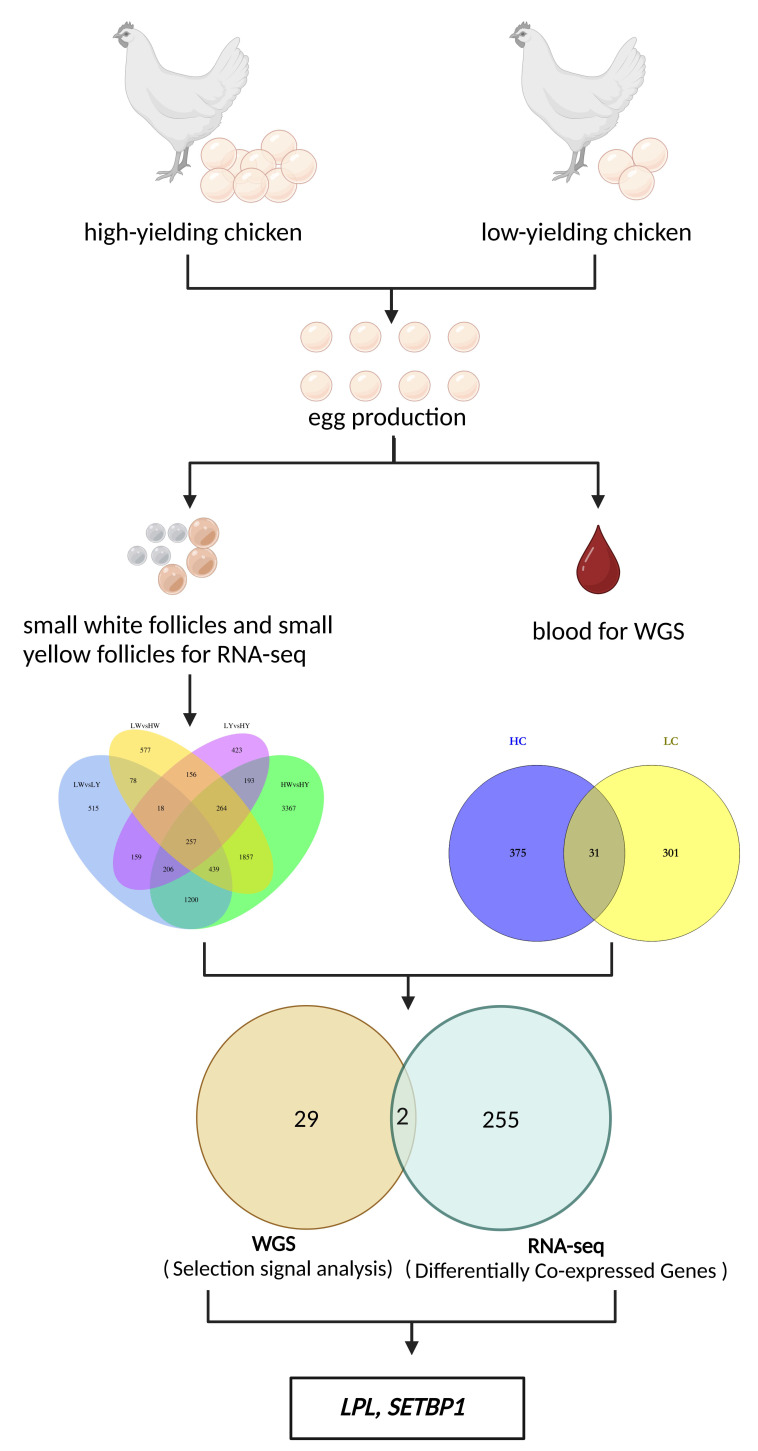
The flow of screening the potential candidate genes for egg production performance in the HC and LC groups (created with BioRender.com).

**Figure 7 ijms-25-09373-f007:**
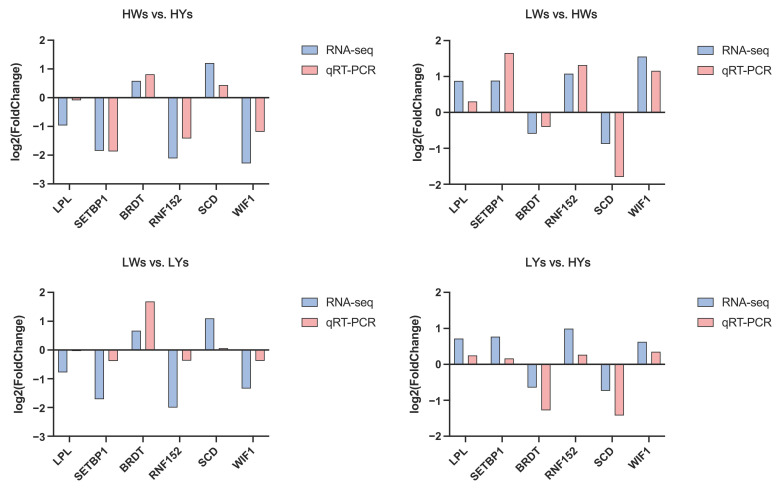
Comparison of mRNA expression levels (log_2_FC) of genes when analyzed by RNA-seq and qRT-PCR. HWs vs. HYs: high-yield chickens’ small white follicles vs. high-yield chickens’ small yellow follicles; LWs vs. HWs: low-yield chickens’ small white follicles vs. high-yield chickens’ small white follicles; LWs vs. LYs: low-yield chickens’ small white follicles vs. low-yield chickens’ small yellow follicles; LYs vs. HYs: low-yield chickens’ small yellow follicles vs. high-yield chickens’ small yellow follicles.

**Table 1 ijms-25-09373-t001:** Primers used in qRT-PCR.

Primers	Accession Number	Forward Primer (5->3)	Reverse Primer (5->3)	Product Length (bp)	Tm (°C)
β-actin [53]	NM_205518.2	CAGCCAGCCATGGATGATGA	ACCAACCATCACACCCTGAT	147	60
LPL	NM_205282.2	ACTGAAACTTTTTCGCCGCTG	TTCATCTCAGCTTCGGGATCG	127	60
SETBP1	XM_046937440.1	TGGCGAGGGATTGAAACCG	GAGATCAGGTCTGCCACCAT	147	60
BRDT	XM_040705075.1	AGGCTGTTCCAGAGGTATCCA	TTGTGACAGTACATACTCTGCTCT	163	60
RNF152	NM_001305034.1	ATGTCGTGCATTTCCAAGCG	CGGCCAAAGTTGCAGTGAAT	146	60
SCD	NM_204890.2	CGGATGCAGACCCTCACAAT	GGGCTTGTAGTATCTCCGCTG	164	60
WIF1	NM_001199607.3	TGTCCTTGCGCTCTTTGGAT	AACCCAACCTGAACCACTGA	103	60

## Data Availability

All data generated or analyzed during this study are included in this published article and its additional files or in the following public repositories. Data have been submitted to a public database under the following accession numbers: whole-genome re-sequencing data [PRJNA1107132] and transcriptome sequencing data [PRJNA1080153].

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
