# Peer review of "Integrating Genomics and Transcriptomics to Identify Candidate Genes for Egg Production in Taihe Black-Bone Silky Fowls (Gallus gallus domesticus Brisson)"

_ijms, 2024, doi:10.3390/ijms25179373_

Round 1
Reviewer 1 Report (New Reviewer)
Comments and Suggestions for Authors
1- LN 12- 14. Considerable variation exists within populations regarding egg-laying performance. This study integrates whole-genome selection signal analysis (SSA) with transcriptome analysis to identify genes associated with egg laying traits in Taihe Black-Bone Silky Fowls.
In my opinion many factors affects variation in egg production may be related to such as nutrition and environment and these factors affects candidate genes. Thus it is highly recommended to add the diets and environmental data
2. Please add the gene bank accession No in table 1 and references.
Author Response
Please see the attachment.

Reviewer 2 Report (New Reviewer)
Comments and Suggestions for Authors
The present research is about “to detect relevant candidate genes and molecular markers influencing egg-laying traits in Taihe Black-Bone Silky Fowls”. This research provides interesting information. However, some important changes need to be made before final publication.
INTRODUCTION
General comments: I recommend increasing this section on the available information on “genomics and transcriptomics”. Also, being clearer in its justification and hypothesis.
Particular comments:
Line 41.- they mention “Eggs constitute the most profitable source of animal protein”. Also, it would be convenient to mention that it is an important alternative due to the “water footprint”. I suggest adding it.
Line 98.- they mention “Transcriptomic studies of these two stages can reveal key genes and pathways that affect the ovulation process and egg production”. Who mentioned this?
Line 112-117.- they mention “Follicles constitute the growth and development environment of oocytes. Follicular surrounding cell types, such as granulosa cells, zona pellucida cells, and cumulus cells, significantly influence oocyte development by secreting a range of hormones and growth factors. Therefore, follicular transcriptome analysis can reveal how follicular cells support and regulate oocyte maturation through changes in gene expression.” Who mentioned this?
Results
General comments: I recommend improving the quality of figures 2, 3, 4, 5, 6.
DISCUSSION In general, I recommend increasing this section. I recommend explaining the following results in more detail:
•“The egg production of HC was higher than that of LC”,
•“The HC and LC groups display distinct distributions of nucleotide diversity, with the HC showing a slightly higher median nucleotide diversity than the LC”
•“KEGG pathway analysis revealed enrichment primarily in pathways related to reproduction, such as Lysosome, MAPK signaling pathway, Calcium signaling pathway, ECM-receptor interaction, and Wnt signaling pathway”
•“Identified two candidate genes, LPL and SETBP1”
MATERIAL AND METHODS
General comments: What animal welfare guidelines were used in this research? I also recommend adding a section for “Statistical analyses.” I recommend specifying the nature of the data, whether they came from a normal distribution, were they transformed? Were there outlier data?
Specific comments:
Line 363-364.- they mention “From the chicken coops of Xichang Fengxiang Poultry Co., Ltd., fifty-four 300-day-old Taihe Black-Bone Silky Fowls with good growth conditions, normal health status, and complete egg-laying records were selected.” I recommend mentioning what the selection, exclusion, management, etc.
Author Response
Please see the attachment.

Reviewer 3 Report (New Reviewer)
Comments and Suggestions for Authors
The paper seems to be lacking in the description of genetic analysis related to egg production overall, although the data obtained through SNP analysis and RNA-seq analysis are considered positive.
- Does Xichang Fengxiang Poultry Co., Ltd raise only one chicken per cage?
- In general, hens that produce fewer eggs tend to be lighter and smaller than hens that produce more eggs. Can weight data be added?
- In line 165, the sentence “the PCA scatter plot reveal distinct clustering patterns between the two groups” looks incorrect. The PCA scatter plot in Figure S1 did not show any distinct cluster. Please add information of statistical analysis.
- All the numbers are too faint to understand. Please improve the quality.
- The two genes (LPL, SETBP1) selected by both WGS and RNA-seq data require precise description of whether they are down-regulated or up-regulated in the HC group.
-delete “Beijing You” between producing and chickens in line 102.
Round 2
Reviewer 1 Report (New Reviewer)
Comments and Suggestions for Authors
Thanks
Author Response
Thank you for your valuable feedback, which has greatly improved the readability and scientific quality of the revised manuscript.
Reviewer 3 Report (New Reviewer)
Comments and Suggestions for Authors
The manuscript is well revised excepting figure qualities. It is very difficult to see figures still. The figures must be increased qualities.
Author Response
Comments 1: The manuscript is well revised excepting figure qualities. It is very difficult to see figures still. The figures must be increased qualities.
Response 1: Thank you for your suggestion. We have already adjusted the image resolution to 600 DPI and ensured that the image quality remains clear when previewed after converting to PDF. However, for reasons we are not entirely sure of, the images become unclear each time they are uploaded to the system. If the issue persists and the images remain unclear, you may want to view the original images or the manuscript in DOCX format that we uploaded to see if they appear clearer. We apologize for any inconvenience this may have caused, and please do not hesitate to contact us if you encounter any further issues.
This manuscript is a resubmission of an earlier submission. The following is a list of the peer review reports and author responses from that submission.
Round 1
Reviewer 1 Report
Comments and Suggestions for Authors
The research included whole genome sequencing and RNA-seq of 54 Taihe Black-Bone Silky Fowl individuals. The idea is very good, but the manuscript requires refinement.
Introduction. This section is inconsistent and should be reconsidered.
Line 121-124 “The aim is to screen for relevant candidate genes and molecular markers influencing egg-laying traits in Taihe black-boned chickens, thereby providing theoretical support for the conservation of chicken genetic resources and breeding efforts.” Conservation of genetic resources and breeding efforts are two mutually exclusive directions. If we want to conduct breeding, we must carry out selection. This means eliminating individuals, which reduces the genetic variability of the population. In the conservation of genetic resources, the goal is to maintain the highest possible genetic variability of a given population, which excludes any selection.
Results.
The evaluation of the results is difficult due to the low resolution of the figures. They need to be improved because they are completely unreadable.
Lines 144-156. The chicken genome is 1.2 Gbp long. The authors identified a total of 315,579,777 SNPs. So, the studied individuals differed in up to 25% of the genome? Is it possible?
Lines 197-204. No abbreviations for HY, HW, LW, LY.
Line 281. “padj”?
Line 301. “Milisits et al.” ad reference number
Material and methods
The methods, tools, software, and reagents are not described with sufficient details to allow another researcher to reproduce the results. Please include information with the names of the chemicals and laboratory equipment used, as well as details regarding the individual stages of the analyses.
When selecting birds for analysis, was the decision based solely on the number of eggs laid, or were pedigree data also considered? Often, the best individuals are more closely related to each other than the average relatedness of the flock. In such cases, analysis based solely on laying performance does not reflect the genetic variability of the population.
The discussion should be expanded.
Reviewer 2 Report
Comments and Suggestions for Authors
1. General comments
This manuscript reports the outcomes of research combining whole genome sequencing and transcriptome analyses to identify the candidate genes or pathways that may play a pivotal role in egg production in Taihe Black-bone silky fowls. Massive next-generation sequencing data generation and integrative bioinformatics analysis were performed. Nonetheless, this study lacks the rationale and methods to integrate whole genome sequencing and transcriptome data. Proper procedures used to integrate them should be included in Materials and Methods.
2. Specific comments
- Have these chickens been under-selected for egg production? If not, a more rigorous statistical analysis should be conducted to identify the selective sweep region. In addition, more specific information needs to be included on experimental animals. Since whole genome sequencing data is available, it is recommended to perform principal component analysis and estimate the inbreeding coefficient of two populations.
- More justification for using the ovarian follicles for transcriptome analysis should be presented. It is not quite sure that ovarian follicles are the proper cells to be investigated to decipher genomic features responsible for differential egg production.
- The image quality of Figures 2 to 5 should be improved for clarity.
